# Post Quantum Design in SPDM for Device Authentication and Key Establishment

**Jiewen Yao ***, **Krystian Matusiewicz and Vincent Zimmer**

Intel Corporation, 2200 Mission College Blvd, Santa Clara, CA 95054, USA
* Correspondence: jiewen.yao@intel.com

**Abstract:** The Security Protocol and Data Model (SPDM) defines a set of flows whose purpose includes the authentication of a computing device's hardware identity. SPDM also allows for the creation of a secure session wherein data communication between two devices has both confidentiality and integrity protection. The present version of SPDM, namely version 1.2, relies upon traditional asymmetric cryptographic algorithms, and these algorithms are known to be vulnerable to quantum attacks. This paper describes the means by which support for post-quantum (PQ) cryptography can be added to the SPDM protocol in order to prepare SPDM for the upcoming world of quantum computing. As part of this paper, we examine the SPDM 1.2 protocol and discuss various aspects of using PQC algorithms, including negotiation of the use of post-quantum cryptography (PQC) algorithms, support for device identity reporting, mechanisms for device authentication, and establishing a secure session. We consider so-called "hybrid modes" where both classical and PQC algorithms are used to achieve security properties, especially given the fact that these modes are important during the transition period from the classical to the quantum computing regime. We also share our experience with implementing a software embodiment of PQC in SPDM, namely "PQ-SPDM", and we provide benchmarks that evaluate a subset of the winning NIST PQC algorithms.

**Keywords:** PQ digital signature; PQ key establishment; post-quantum SPDM; device authentication; device secure session

## 1. Introduction

### 1.1. The Threat of Quantum Computers

Virtually all asymmetric cryptosystems used in practice are based on one of two difficult computational problems, either integer factorization or finding discrete logarithms in appropriately chosen cyclic groups. Concrete examples include RSA, ECDSA, and EdDSA for digital signatures and ephemeral DiffieHellman over finite fields (FFDHE) or elliptic curves (ECDHE) for key establishment.

Unfortunately, all of the above listed algorithms can be broken by a sufficiently large quantum computer. This lack of safety is based upon attacks leveraging Shor's algorithm [1–4] that can break cryptosystems relying upon the hardness of integer factorization and discrete logarithms in abelian groups. While the race to build large-scale quantum computers is still at an early phase, a recent survey [5] shows that many experts expect quantum computers capable of breaking RSA2048 to be available in fifteen to thirty years.

To address this challenge, alternatives to vulnerable algorithms have been studied over the past fifteen years, and the notion of *post-quantum cryptography algorithms* (PQC) has emerged. They are classical algorithms (requiring only classical computers to be executed) that are expected to be secure against both classical and quantum attacks. This process accelerated after the National Institute of Standards and Technology (NIST) announced a competition to select a suite of quantum resistant algorithms [6].

### 1.2. Device Security and SPDM

The Security Protocol and Data Model (SPDM) 1.2 specification [7] is defined by the Distributed Management Task Force (DMTF). SPDM entails various usages, including device identity collection, device authentication, measurement collection, and device secure session establishment. The SPDM protocol is a standard for use by the broad device community. SPDM has been adopted by various other standard groups, including Peripheral Component Interconnect (PCI) [8], Compute Express Link (CXL) [9], Mobile Industry Processor Interface (MIPI) [10], and Trusted Computing Group (TCG) [11]. For example, various application protocols have been created above the SPDM protocol, including PCI express (PCIe) Integrity and Data Encryption (IDE) Key Management (IDE_KM) protocol [12] for PCIe link encryption, CXL Integrity and Data Encryption (IDE) Key Management (IDE_KM) protocol [13] for CXL link encryption, PCI express TEE Device Interface Security Protocol (TDISP) [14] for confidential computing, and MIPI Service Association Configuration Protocol (SACP) [15] for automotive system security management. The TCG Platform Firmware Profile Standard [16] and Open Compute Project (OCP) Attestation of System Components requirements [17] also relies upon SPDM-based device attestation for the associated the platform.

The current SPDM 1.2 specification uses traditional asymmetric cryptographic algorithms that will be vulnerable to quantum attacks. Since some devices are expected to have long lifetimes in the field and need to be supported well beyond 2030, it seems prudent to consider supporting post-quantum cryptography (PQC) in the SPDM protocol.

A significant body of research has been dedicated to the augmentation of network security infrastructure with PQC capabilities, including X.509 certificates [18–22], Media Access Control security (MACsec) [23], Internet Protocol Security (IPSec) [24,25], Transport Layer Security (TLS) [26–36], Secure Shell (SSH) [32,37], and WireGuard [38–40]. In addition, PQC algorithms have been implemented in devices as part of their secure boot capability [41,42].

As far as we know, this paper is the first study of PQC algorithm adoption for a *device* communication security protocol, such as SPDM. This aspect seems to be currently particularly relevant for the Internet of Things (IoT) ecosystem. The IoT market is expanding with diverse security needs, including long-term security requirements. In addition, these IoT devices need to be supported by backend services where performance is also critical due to the sheer number of endpoint devices.

### 1.3. Our Contributions

The focus of this analysis includes infrastructure to support post-quantum cryptographic algorithms in the SPDM protocol, including algorithm negotiation, device certificate transfer, digital signature signing and verification, and authenticated key exchange for secure sessions. We evaluated a prototype implementation of a PQC-enlightened SPDM stack in a device with different modes:

- Traditional mode: PQC algorithms are not used.
- PQC mode: only PQC algorithms are used.
- Hybrid mode: both traditional algorithms and PQC algorithms are used.

Hybrid mode was studied in [43,44] and recommended by NIST during the transition and migration phases [45].

We also consider design challenges for adopting post-quantum algorithms for devices with SPDM capability, including considerations for backward compatibility and message size.

The paper is organized as follows. We begin with a brief recap of SPDM in Section 2. Section 3 is dedicated to the extending algorithm negotiation phase. Then, we follow with changes to support PQC-ready device identities in Section 4, and we continue with device authentication in Section 5. The next topic entails establishment of secure session keys, which is treated in detail in Section 6 with an associated security analysis. Finally, we report

the results of our proof-of-concept implementation and related discussion in Section 7 and Section 8, respectively.

The PQC algorithm selection has been driven by the National Institute of Standards and Technology (NIST) [46]. Although the protocol design is algorithm agnostic, thus allowing usage with any new PQC scheme, we report performance results for the four recently announced winners of the NIST PQC competition.

As a final note, given that symmetric cryptographic algorithms can be used in the quantum computing world with increased key sizes and larger hash digests, we do not study any symmetric cryptographic algorithms for SPDM, including hash algorithms or Authenticated Encryption with Associated Data (AEAD) algorithms.

### 1.4. Related Work

Various studies have been undertaken to investigate options for adding PQC to devices. For example, ref. [47] provided a general survey for PQC algorithm adoption in an IoT device; ref. [48,49] evaluated the runtime PQC algorithm performance of the TLS protocol for an IoT or embedded devices. In [50], the authors evaluated the boot time performance impact of PQC digital signature verification for secure boot; refs. [51,52] provided a survey of a hardware implementation of Lattice-Based Cryptography (LBC) for IoT Devices. In [53], the authors discussed the usage of LBC, including identity-based encryption, homomorphic encryption, and secure authentication key exchange, along with the performance impact.

## 2. SPDM Background

### 2.1. SPDM Device Authentication

In the SPDM specification, the authentication of the identity of a device involves two steps: device identification and device authentication.

During the device manufacturing phase, each device is provisioned with a device certificate chain. This certificate chain can be treated as the device identity. The device certificate chain includes all of the certificates from a trusted root certificate authority (CA) certificate, and these certificates chains are linked to a device-specific leaf certificate. The device certificate contains the public key that corresponds to the device private key. The root CA endorses the device public/private key pair through the certificate chain. At runtime, an SPDM initiator, also known as the "SPDM requester", may use a `GET_-CERTIFICATE` message to ask an SPDM device, also known as an "SPDM responder", to return its certificate chain as the device's identity.

The SPDM device authentication step uses a challenge–response mechanism. The initiator (requester) sends a `CHALLENGE` message to the SPDM device (responder). The `CHALLENGE` message includes a nonce to prevent any replay attacks. As a next step, the device signs the challenge with its private key and returns a `CHALLENGE_AUTH` message. The authentication initiator, in turn, can verify the digital signature by using the device public key extracted from the certificate chain.

### 2.2. SPDM Secure Session

The SPDM specification also allows for two devices to establish a secure communication channel, similar to the capability of the network protocol Transport Layer Security (TLS) 1.3. To initiate such a channel, an SPDM requester and an SPDM responder may use an authenticated key exchange protocol to derive a set of session keys. These session keys will provide for both confidentiality and integrity of the communication data by using encryption and message authentication.

### 2.3. SPDM Algorithm Negotiation

The SPDM specification defines a message `NEGOTIATE_ALGORITHMS`, which is sent by an SPDM requester, along with a message `ALGORITHMS`, which in turn is returned by the SPDM responder. Together these two messages perform the task of negotiating a common set of cryptographic algorithms. The cryptographic algorithm selection in SPDM is different

from the implementation mechanism in TLS. Specifically, a TLS entity selects one cipher suite that covers all algorithms, including key exchange or agreement, authentication, block or stream ciphers, and message authentication. An SPDM requester, on the other hand, lists all individual algorithms it can support, such as hash algorithms, responder direction asymmetric digital signature algorithms, requester direction asymmetric digital signature algorithms, key exchange algorithms and Authenticated Encryption with Associated Data (AEAD) ciphers. With this requester information in hand, the SPDM responder chooses one of these options in each category as the final negotiated algorithm. This differs from TLS cipher suites in that each type of SPDM algorithm can be negotiated separately.

## 3. Post-Quantum Design for SPDM Algorithm Negotiation

The SPDM specification defines cryptographic algorithms separately. An SPDM entity can negotiate an individual cipher, such as a hash algorithm, a responder direction asymmetric digital signature algorithm, a requester direction asymmetric digital signature algorithm, a key exchange algorithm, and an Authenticated Encryption with Associated Data (AEAD) cipher suite. Table 1 shows an example.

**Table 1.** Current SPDM algorithm negotiation (example).

| Algorithm | Requester's List | Responder's Selection |
|---|---|---|
| Hash | SHA256,<br>SHA384 | SHA384 |
| Responder Digital Signature | RSASSA_3072,<br>ECDSA_NIST_P256,<br>ECDSA_NIST_P384 | ECDSA_NIST_P384 |
| Requester Digital Signature | RSASSA_3072,<br>RSAPSS_3072 | RSAPSS_3072 |
| Key Exchange | FFDHE_3072,<br>ECDHE_secp256r1,<br>ECDHE_secp384r1 | ECDHE_secp384r1 |
| AEAD | AES_256_GCM,<br>CHACHA20_POLY1305 | AES_256_GCM |

In order to support a post-quantum world, we extend three algorithm types: PQC responder digital signature algorithm, PQC requester digital signature algorithm, and PQC key exchange algorithm. The original responder digital signature algorithm, requester digital signature algorithm, and key exchange algorithms are interpreted as traditional algorithms only for purposes of compatibility support.

Table 2 shows an example of an SPDM algorithm negotiation with PQC in traditional mode, PQC mode, and hybrid mode. The Responder Digital Signature designates the SPDM asymmetric key signature algorithm for purposes of signature generation by the responder and verification by the requester. The Requester Digital Signature describes the SPDM asymmetric key signature algorithm for signature generation by the requester and verification by the responder. The PQC algorithms are separated from traditional algorithms because we expect that the responder will select both one of the PQC algorithms and one of the traditional algorithms in hybrid mode.

Assuming that the SPDM Requester has full capabilities, including traditional-only, PQC-only and hybrid mode, the requester needs to declare all supported algorithms. Given this declaration, the SPDM responder needs to make the final decision. As such, if the responder wants to choose traditional-only mode, it will select one of the traditional algorithms and not select any PQC algorithm. In turn, if the responder wants to choose PQC-only mode, it will select one of the PQC algorithms and not select any of the traditional algorithms. Finally, if the Responder wants to choose hybrid mode, it will select one traditional algorithm and one PQC algorithm.

**Table 2.** SPDM Algorithm negotiation with PQC (example).

| Algorithm | Requester's List (Full Capability) | Responder's Selection (Trad. Mode) | Responder's Selection (PQC Mode) | Responder's Selection (Hybrid Mode) |
|---|---|---|---|---|
| Hash | SHA256, SHA384 | SHA384 | SHA384 | SHA384 |
| Responder Digital Signature | RSASSA_3072, ECDSA_NIST_P256, ECDSA_NIST_P384 | ECDSA_NIST_P384 | - | ECDSA_NIST_P384 |
| Requester Digital Signature | RSASSA_3072, RSAPSS_3072 | RSAPSS_3072 | - | RSAPSS_3072 |
| Key Exchange | FFDHE_3072, ECDHE_secp256r1, ECDHE_secp384r1 | ECDHE_secp384r1 | - | ECDHE_secp384r1 |
| AEAD | AES_256_GCM | AES_256_GCM | AES_256_GCM | AES_256_GCM |
| PQC Responder Digital Signature | Falcon-512, Falcon-1024 | - | Falcon-512 | Falcon-512 |
| PQC Requester Digital Signature | Dilithium2, Dilithium5 | - | Dilithium2 | Dilithium2 |
| PQC Key Exchange | Kyber512, SIDH-p434, SIKE-p434 | - | SIDH-p434 | SIDH-p434 |

There is one limitation for the use of this mechanism, though. Namely, a requester or a responder may want to say: "I can support either traditional mode or PQC mode, but I do not want to support hybrid mode". For this case, there is no way to pass such information via the NEGOTIATE_ALGORITHMS request and ALGORITHMS response messages. As such, we need to use the GET_CAPABILITIES request and CAPABILITIES response messages. To accommodate this use-case, we have added additional capabilities bits, including:

- PQC Capability: this means an entity supports PQC mode;
- Hybrid Capability: this means an entity support hybrid mode.

If both entities indicate PQC capability, then PQC mode is chosen. If both entities indicate hybrid capability, then hybrid mode is chosen. If both PQC capability and hybrid capability are selected, then hybrid mode takes precedence. We do not define a traditional algorithm capability for compatibility consideration, though.

The SPDM specification also defines the timing requirements. The requester and the responder need to inform their peer of a CTExponent timing parameter, which is the maximum amount of time for cryptographic processing. A PQC algorithm might have different timing requirements compared to a traditional algorithm. In addition, the timing in hybrid mode could be the addition of timing for both traditional mode and PQC only mode. Unfortunately, the timing parameter in CAPABILITIES is negotiated before the algorithm negotiation in ALGORITHMS. As such, it is difficult to use a single CTExponent to indicate the timing parameter of three different modes. To address this concern, the setting of a maximum value is one option, but this maximum value might cause the requester to wait for en excessive period prior to a retry. The other option is to add a new PQCTExponent to indicate the PQC-specific timing. With this new parameter in hand, the final interpretation of the timing parameter could be:

- Timing in Traditional Mode: $2^{\texttt{CTExponent}}$;
- Timing in PQC Mode: $2^{\texttt{PQCTExponent}}$;
- Timing in Hybrid Mode: $2^{\texttt{CTExponent}} + 2^{\texttt{PQCTExponent}}$.

*3.1. Design Considerations*

3.1.1. Mode Identification

A requester and a responder may negotiate traditional algorithms and PQC algorithms separately. A device that only supports traditional mode may provide or select traditional algorithms and ignore PQC algorithms. A PQC-mode-only device may provide or select PQC algorithms and ignore traditional algorithms. Finally, a hybrid mode-aware device may provide a list of traditional algorithms and a list of PQC algorithms. Then, the hybrid mode-aware device peer may select one traditional algorithm and one PQC algorithm. For any of these cases, an SPDM requester or an SPDM responder can ascertain the final negotiated mode by checking if both traditional algorithms and PQC algorithms have been negotiated.

The other option is to merge the traditional algorithm and PQC algorithm together in one field. There is no impact to a requester in choosing this approach. The impact for a responder is that we need to allow the algorithm selection field to designate one or two algorithms. If only one algorithm is selected, then it is traditional mode or PQC mode. If two algorithms are selected, namely one that is traditional and the other that is PQC, then it is hybrid mode.

3.1.2. No Combinatorial Explosion

Since SPDM allows both entities to negotiate algorithms individually, there is no danger of a combinatorial explosion of algorithms. We use a similar technique to handle the hybrid mode. To support this economy of storage, SPDM uses one bit to indicate one algorithm. As such, in order to support $N$ traditional mode ciphers and $M$ PQC mode ciphers, the total required bit space is $(N + M)$.

3.1.3. Security Level Matching

Technically, traditional algorithms and PQC algorithms are orthogonal. In the real world, a product may need to meet an NIST security level requirement. As such, an entity should choose the required traditional algorithm and a PQC algorithm with a matching security level in hybrid mode. For example, the Level 1 PQC algorithm should go with the ECC P-256 curve, the Level 3 PQC algorithm should go with the ECC P-384 curve, and the Level 5 PQC algorithm should go with the ECC P-521 curve.

3.1.4. Algorithm Set Matching

In addition to a set of NIST algorithms, such as SHA, AES, RSA, DH, and ECC, SPDM 1.2 also adds support for Shang-Mi (SM) algorithms, such as SM3, SM4, and SM2. In the real world, the product may choose either an NIST algorithm set or an SM algorithm set. In the future, if NIST or SM define their own PQC algorithm sets, the PQC algorithm should go with the traditional algorithm in the same set.

3.1.5. Hash Algorithm Used in Digital Signature

Some traditional digital signature algorithms require fixed length input messages for signing. To reduce the combinatorial explosion problem, the hash algorithm is negotiated separately from the digital signature algorithm in SPDM. Some PQC digital signature algorithms, such as SPHINCS+, have an associated hash algorithm such as SHA256 or SHAKE256. This leads to the challenge of how to indicate which hash algorithm is required for a given PQC digital signature. Currently, we let the original SPDM hash algorithm only associate with the traditional digital signature. The PQC digital signature algorithm should indicate the hash algorithm explicitly, though. For example, `SPHINCS+-SHA256-128f-robust` or `SPHINCS+-SHAKE256-128f-robust` are potential options.

**4. Post-Quantum Design for SPDM Device Identity**

The SPDM specification uses an X.509 certificate chain as the device identity. If an SPDM device supports a PQC algorithm, whether it be PQC and/or hybrid mode, the

device should carry an X.509 certificate with PQC support. This X.509 certificate can be a PQC or hybrid X.509 certificate. In hybrid mode, the SPDM requester can use the `GET_-CERTIFICATE` command to retrieve one X.509 certificate chain and the `SET_CERTIFICATE` command to provision one. This one certificate chain contains both a traditional algorithm and a PQC algorithm. In either PQC mode or traditional mode, this hybrid certificate chain may still be used since the verifier can just skip the unneeded algorithm and certificate digital signature.

An alternate approach is to extend the `GET_CERTIFICATE` command to indicate which type of certificate chain is required, namely a traditional certificate chain or a PQC certificate chain. The Certificate Type can be in `param2` and includes

- Traditional Certificate Chain (0),
- PQC Certificate Chain (1),
- Hybrid Certificate Chain (2).

In hybrid mode, the requester needs to request the certificate chain twice: one for the traditional certificate chain and the other for the PQC certificate chain.

The X.509 certificate format is out of scope of the SPDM specification. As such, we will not discuss the details here.

The SPDM specification may also allow a device to provision a raw public key of the peer during the manufacturing phase. In this case, the trust of the public key of the peer is established without the certificate-based public key infrastructure. If PQC is required in the raw public key scenario, then the manufacturer may need to provision the public keys for both a traditional algorithm and a PQC algorithm. The format of the raw public keys is out of scope of the SPDM specification. Given that the raw key format is implementation specific, we omit a discussion of the details here.

The SPDM 1.2 specification defines an alias certificate mode to support device compatibility with the Trusted Computing Group (TCG) Device Identity Composition Engine (DICE) specification [54]. A DICE device may include a device ID certificate and an alias certificate. The device ID certificate includes the device ID key derived from the Compound Device Identity (CDI) value computed by the DICE process. The certificate depends upon the unique device secret (UDS) and measurement of DICE layer 0. The alias certificate includes a device alias key that is computed using the last CDI value in the chain of the device Trust Computing Base (TCB) component. The alias certificate is usually generated during the runtime phase of a device because the CDI includes the measurement of the last layer, which includes mutual firmware elements. If a DICE device chooses to support a PQC alias certificate chain or hybrid alias certificate chain, each DICE layer needs to support the PQC algorithm.

*4.1. Design Considerations*

4.1.1. No Duplicate Message in Transport

In hybrid mode, if we require an entity to pass only one hybrid certificate chain, then the message includes the identity information. In this case, the authority information just needs to be transmitted once.

If we require an entity to pass both a traditional certificate chain and a PQC certificate chain, then the identity information and authority information will be transmitted twice. In addition, both certificate chains need to be included in the transcript calculation. This doubling of communication is not efficient and might cause latency problems. In addition, the entity needs to check if the two certificate chains use the same identity information or authority information. If the information provided is different, then the entity needs to record the difference and verify against the pre-defined policy. Such bookkeeping adds significant complexity.

### 4.1.2. Local Storage Size

When an entity considers the compatibility requirements, the entity will consider the capability of the peer. Because the peer may support traditional mode, PQC mode, and hybrid mode, the entity may need to prepare a traditional mode certificate chain, a PQC mode certificate chain, and a hybrid mode certificate chain. A small device may only have limited storage size. As such, provisioning three certificate chains might not be the best choice.

### 4.1.3. Transport Message Size

The SPDM specification does not define the message size limitation. However, any secured message using SPDM [55] is constrained to a 16-bit application data length field. As such, the maximum size of an encrypted message is $2^{16}$ bytes. The SPDM transport layer binding specifications also have a size limitation. For example, the PCI Data Object Exchange (DOE) mailbox [12] uses an 18-bit length field for the double-word (4 byte) number, which means the maximum message size is $2^{20}$ bytes. The Management Component Transport Protocol (MCTP) over System Management Bus (SMBus)/Inter-Integrated Circuit (I2C) [56–58] only supports 256 bytes as a maximum size in a given packet.

For a large message that may potentially exceed the 256-byte limitation, SPDM 1.0 and 1.1 defines a command-specific chunking mechanism. For example, the `GET_CERTI-FIACTE` request message includes both 16-bit `OFFSET` and 16-bit `LENGTH` fields to indicate the requested certificate data buffer offset and length. The `CERTIFICATE` response message includes a 16-bit `PortionLength` and 16-bit `RemainderLength` to indicate the transmitted data buffer length and remaining data buffer length, respectively. SPDM does not rely on any low level transport layer fragmentation support. A more generic chunking mechanism (`CHUNK_CAP`) is defined in SPDM 1.2. As such, all SPDM messages can support chunking after the chunking capability is negotiated, such as the `SET_CERTIFIACTE` and `CSR`.

The SPDM public certificate size depends upon the PQC public key size. Some PQC algorithms, such as Dilithium and Falcon, use public keys larger than 256 bytes. The public key sizes of both Rainbow and GeMSS algorithms are larger than $2^{16}$ bytes. As such, the current `GET_CERTIFICATE` command cannot meet the requirement. We may need to extend the 16-bit offset and length field to 32 bits in a new version of `GET_CERTIFICATE`, or we can redefine `GET_CERTIFICATE` to use the generic SPDM 1.2 chunking mechanism. However, considering a recent attack [59] on Rainbow, these PQC algorithms with larger key size might not be a concern going forward.

In addition, we may define a new format for secured message using SPDM [55] by extending the 16-bit application length field to 32 bits. This would help accommodate devices without the capability for chunking.

## 5. Post-Quantum Design for SPDM Device Authentication

Some SPDM messages, such as `CHALLENGE_AUTH` response, `MEASUREMENTS` response, `KEY_EXCHANGE_RSP` response, and `FINISH` request, require a digital signature. The digital signature can include RSA, ECDSA, EdDSA, and SM2. The SPDM specification defines the binary format of the digital signature for a specific algorithm with a fixed size.

The signature format in PQC mode is exactly the same as the one in traditional mode. The signature size can be interpreted as the digital signature size of the PQC algorithm. In hybrid mode, we require that both a traditional algorithm and a PQC algorithm sign the same message data. The digital signature field in the SPDM message entails the concatenation of the two signatures. The first part is the traditional digital signature with fixed size, and the second part is the PQC digital signature. In order to maintain compatibility, we do not add the signature size field. The format of the PQC digital signature should be defined by the PQC algorithm.

Figure 1 shows the high-level view of SPDM authentication message flow.

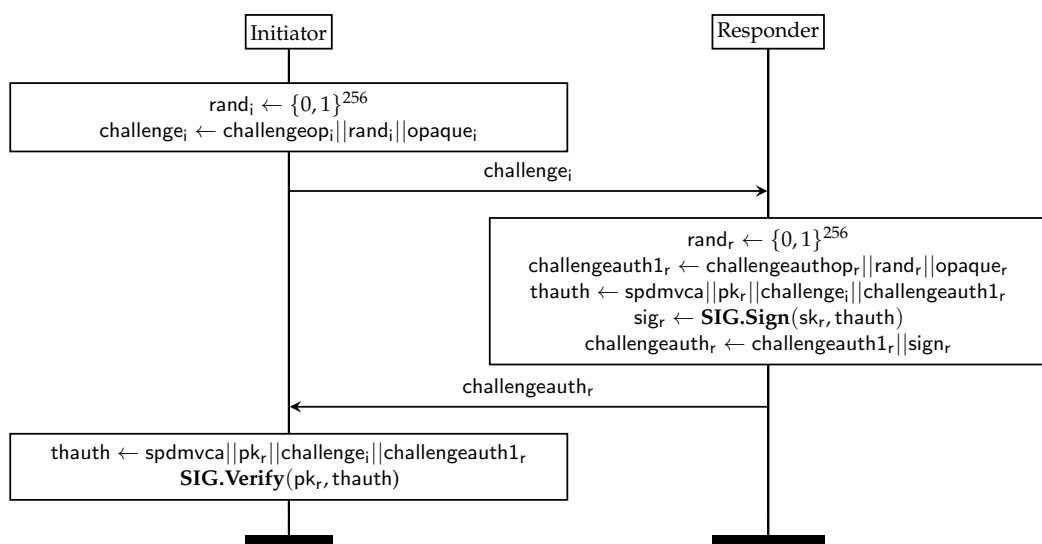

**Figure 1.** SPDM device authentication flow.

*5.1. Message Flow*

5.1.1. First Message

> CHALLENGE (initiator to responder)

1. The initiator generates a 256-bit random number ($rand_i$) as a challenge value to prevent the replay attacks.
2. The CHALLENGE message ($challenge_i$) is the concatenation of the opcode ($challengeop_i$), the random number ($rand_i$), and the initiator specific opaque data.

5.1.2. Second Message

> CHALLENGE_AUTH (responder to initiator)

1. The responder also generates a 256-bit random number ($rand_r$).
2. The responder can prepare the CHALLENGE_AUTH message ($challengeauth1_r$), which is concatenation of the opcode ($challengeauthop_r$), the random number ($rand_r$), and the responder specific opaque data.
3. Now, the responder creates the transcript for authentication ($thauth$), which is the concatenation of the negotiated SPDM protocol version, capability and algorithm ($spdmvca$), the permanent public certificate chain or public key of the responder as identity information ($pk_r$), the CHALLENGE message from initiator ($challenge_i$), and the response generated CHALLENGE_AUTH message ($challengeauth1_r$).
4. The next step is to sign the transcript ($thauth$) with the responder secret key ($sk_r$) and generate the signature ($sig_r$).
5. The final full CHALLENGE_AUTH message ($challengeauth_r$) is the concatenation of the response generated CHALLENGE_AUTH message ($challengeauth1_r$) and the signature ($sig_r$).

After the initiator receives the CHALLENGF_AUTH, it will follow the same process to construct the transcript for authentication ($thauth$) and verify the signature ($sig_r$) with the responder public key ($pk_r$). If the signature verification passes, then the initiator authenticates the responder successfully.

*5.2. Design Considerations*

5.2.1. Hybrid Digital Signature

For a hybrid digital signature, we require that both a traditional algorithm and a PQC algorithm be used to sign the same message data. After signing, two signatures are concatenated together.

We do not choose the option to let one algorithm sign the digital signature output from the other algorithm. This aligns with the current practice of TLS hybrid mode.

### 5.2.2. No Duplicate Message in Transport

In hybrid mode, both the traditional and PQC digital signatures are combined together. There is no need to send the `CHALLENGE/CHALLENGE_AUTH` twice, namely one for classic mode and the other for PQC mode. Instead, the transcript calculation just needs to happen one time.

### 5.2.3. Transport Message Size

As we discussed in a previous section, there is a size limitation in the transport message. Unfortunately, most post-quantum algorithms have a large digital signature size. The signature size of Dilithium, Falcon, SPHINCS+, and picnic algorithms are larger than 256 bytes. The signature size of `picnic_{L3,L5}_{FS,UR,full}` can be larger than $2^{16}$ bytes.

Currently, there are multiple SPDM messages that include a digital signature, including `CHALLENGE_AUTH` response, `MEASUREMENTS` response, `KEY_EXCHANGE_RSP` response, and `FINISH` request message. In order to support the transportation of these large messages, we can use the generic SPDM 1.2 chunking message for these instances.

### 5.2.4. Timing

As we discussed in a previous section, a PQC algorithm may have a different signature generation timing requirement. Some hash-based signature (HBS) algorithms, such as SPHINCS+, are much slower than the lattice-based algorithms, such as Dilithium2 or Falcon. As such, the `PQCTExponent` should be larger than the maximum signing time for the supported PQC signature algorithms.

## 6. Post-Quantum Design for SPDM Secure Session

SPDM defines two ways to build a secure session based on either a pre-shared key (PSK) or an asymmetric key exchange. The PSK-based key exchange involves only a hash-based message authentication code (HMAC), which is still secure against quantum adversaries. The asymmetric key exchange uses an ephemeral Diffie–Hellman (DH) over finite fields or elliptic curves. The SPDM specification defines the binary format of the DH public key (`ExchangeData`) for both the SPDM requester and the SPDM responder.

The SPDM 1.2 specification only supports FFDHE and ECDHE. One-way authentication is always required in an SPDM key exchange, whereas mutual authentication is optional. Figure 2 provides the high-level overview of SPDM asymmetric handshake message flow for mutual authentication. The step with an asterisk in the FINISH message is not required for one-way authentication. The DH-related action is highlighted in blue. The DH ephemeral secret key (*esk*) is highlighted in red, and the DH ephemeral public key (*epk*) is highlighted in green.

NIST currently supports Key Encapsulation Mechanism (KEM) for the shared key generation. The only DH-style PQC, namely Supersingular Isogeny Diffie–Hellman (SIDH), has been proven vulnerable recently [60]. We adopted a KEM-style algorithm to support the `ExchangeData` format in PQC mode. This is similar to the approach in traditional mode. In this case, the requester's `ExchangeData` is the requester's public key generated using the PQC KEM key generation algorithm. The responder's `ExchangeData` is the responder's cipher text generated using the PQC KEM encapsulation. In hybrid mode, the `ExchangeData` field in the SPDM message should be the concatenation of the two key `ExchangeData` fields. The first part is the traditional key `ExchangeData` with a fixed size, and the second part is the PQC key `ExchangeData` with a fixed size. The format of the PQC key `ExchangeData`, such as public key and cipher text, should be determined by the selected PQC algorithm. With this approach, only the key agreement part of SPDM is converted to PQC. The remaining portion, such as the identity authentication, is unchanged.

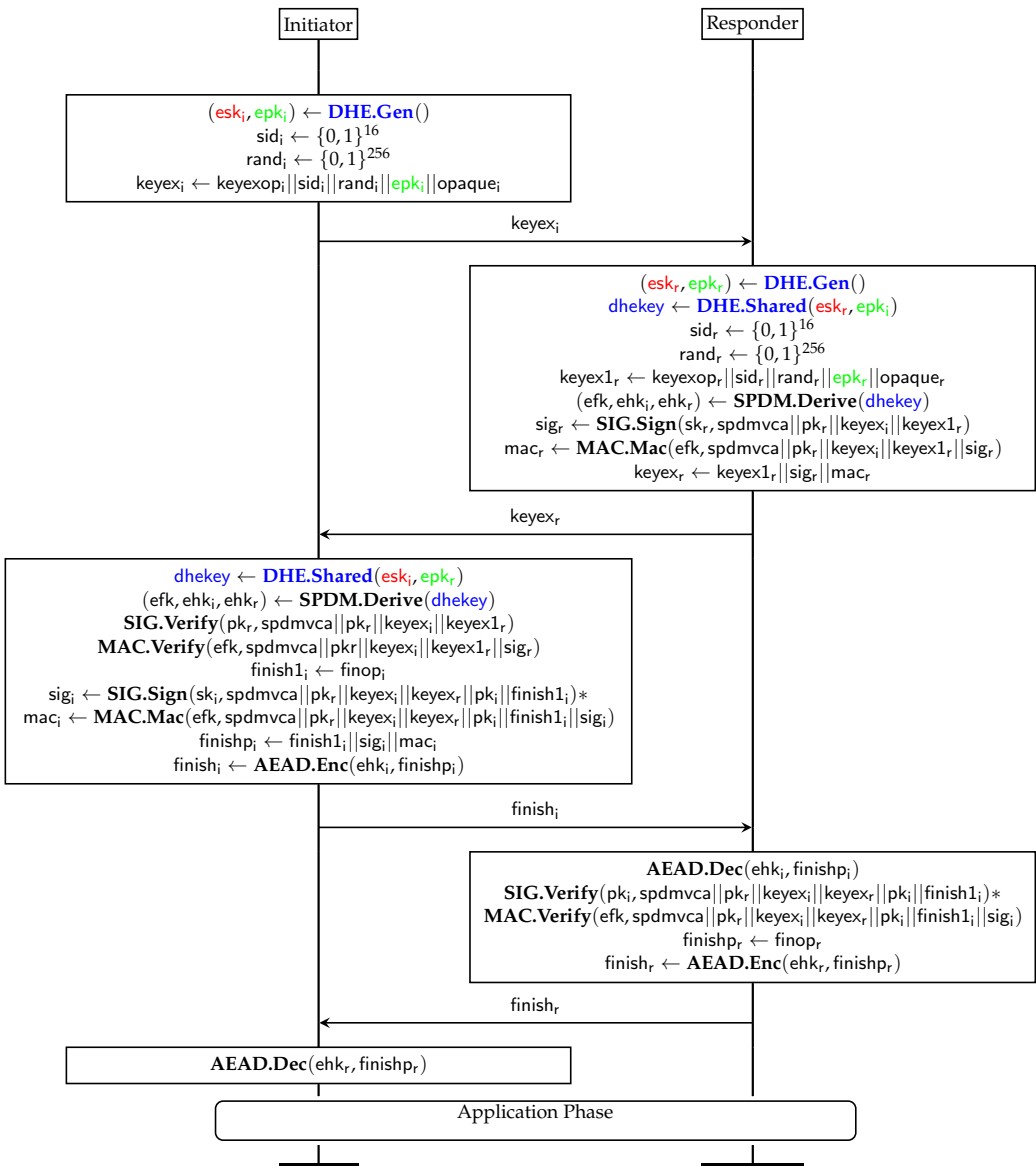

**Figure 2.** Traditional SPDM DHE-based key exchange flow.

We did notice that the KEM-based authentication [61] was adopted by several standard proposals, such as WireGuard [38] and KEMTLS [27,35,36]. However, following this approach for SPDM will entail a significant impact for the SPDM `KEY_EXCHANGE` message flow, as well as other digital-signature-based commands such as SPDM `CHALLENGE` or `GET_-MEASURMENT`. Our future work entails exploring these various design approaches to support SPDM KEM-based authentication, including analysis of any potential benefit therein.

### 6.1. SPDM KEM Message Flow

For PQC-SPDM, we use a KEM-based authenticated key exchange to replace DHE. Figure 3 provides the high-level overview of a PQ-SPDM KEM-based handshake message flow with mutual authentication. The step in the FINISH message that is typeset with an asterisk is not required in one-way authentication. The KEM-related action is highlighted in blue. The KEM ephemeral secret key (*esk*) is highlighted in red, and the KEM ephemeral public key (*epk*) and cipher text are each highlighted in green. Here, *spdmvca* means the SPDM transcript for the SPDM command/response: `GET_VERSION/VERSION`, `GET_-CAPABILITIES/CAPABILITIES`, and `NEGOTIATE_ALGORITHMS/ALGORITHMS`.

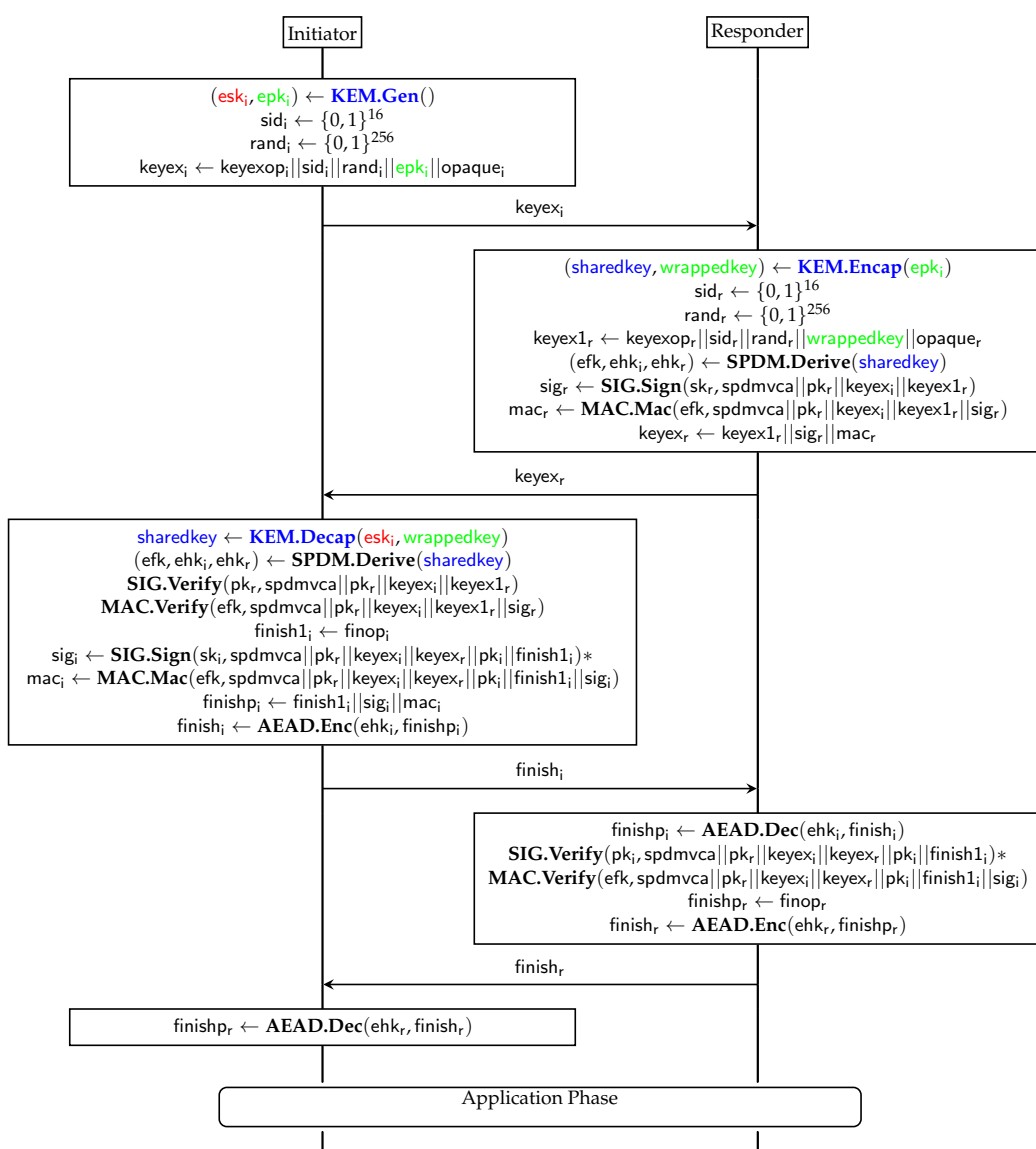

**Figure 3.** PQ-SPDM KEM-based key exchange flow

### 6.1.1. First Message

KEY_EXCHANGE (initiator to responder)

1.  The initiator needs to use the KEM algorithm to generate an ephemeral private/public key pair ($esk_i$ and $epk_i$).
2.  To identify the session, the initiator generates a 16-bit session ID ($sid_i$) as the first half of a 32-bit session ID.
3.  To prevent the replay attacks, the initiator generates a 256-bit random number ($rand_i$).
4.  The KEY_EXCHANGE message ($keyex_i$) is the concatenation of the opcode ($keyexop_i$), the initiator generated session ID ($sid_i$), the random number ($rand_i$), the ephemeral public key ($epk_i$), and the initiator specific opaque data.

### 6.1.2. Second Message

KEY_EXCHANGE_RSP (responder to initiator)

1.  After the responder receives the KEY_EXCHANGE message, the responder uses the KEM algorithm to encapsulate the ephemeral public key ($epk_i$) and derives the shared session key and the cipher text.

2.  The responder generates a 16-bit session ID ($sid_r$) as the second half of the 32-bit session ID. The final session ID is the concatenation of the initiator's session ID and the responder's session ID ($sid_i||sid_r$).

3.  The responder also generates a 256-bit random number ($rand_r$).

4.  Now, the responder can prepare the KEY_EXCHANGE_RSP message ($keyex1_r$), which is concatenation of the opcode ($keyexop_r$), the half of the session ID ($sid_r$), the random number ($rand_r$), the KEM cipher text, and the responder specific opaque data.

5.  From the shared key, the responder uses the SPDM-defined key schedule algorithm to derive the ephemeral finish key ($efk$), initiator direction ephemeral handshake key ($ehk_i$), and responder direction ephemeral handshake key ($ehk_r$). The ephemeral finish key ($efk$) will be used to generate a message authentication code (MAC) for the transcript. The ephemeral handshake key will be used to AEAD the rest of messages in the handshake phase such as FINISH and FINISH_RSP.

6.  In order to prevent man-in-the-middle-attacks, the responder will create a transcript, sign the transcript with its permanent private key ($sk_r$) ,and generate the digital signature ($sig_r$). Per the SPDM specification, the transcript of KEY_EXCHANGE_RSP for signing is the concatenation of the negotiated SPDM protocol version, capability and algorithm ($spdmvca$), the permanent public certificate chain or public key of the responder as identity information ($pk_r$), the KEY_EXCHANGE message from initiator ($keyex_i$), and the response generated KEY_EXCHANGER_RSP message ($keyex1_r$).

7.  The next step is to create another transcript and MAC the transcript with the ephemeral finish key ($efk$) and generate the MAC ($mac_r$). Per the SPDM specification, the transcript of KEY_EXCHANGE_RSP for MAC is the concatenation of the transcript of KEY_EX-CHANGE_RSP for signing and the digital signature ($sig_r$).

8.  The final full KEY_EXCHANGE_RSP message ($keyex_r$) is the concatenation of the response generated KEY_EXCHANGER_RSP message ($keyex1_r$), the digital signature ($sig_r$), and the MAC ($mac_r$).

### 6.1.3. Third Message

FINISH (initiator to responder)

1.  Once the initiator receives the KEY_EXCHANGE_RSP message, it can use the KEM algorithm to decapsulate the cipher text with its ephemeral private key ($esk_i$) and generate the shared key.

2.  The initiator can follow the SPDM-defined key schedule algorithm to derive the ephemeral finish key ($efk$) and initiate the direction ephemeral handshake key ($ehk_i$) and the responder direction ephemeral handshake key ($ehk_r$). These keys should be same as the one derived by the responder.

3.  Then, the requester will follow the same process to construct the transcript of KEY_EX-CHANGE_RSP for signing and verifying the digital signature ($sig_r$) with the responder's permanent public key ($pk_r$). If the digital signature verification fails, then the initiator will terminate the session handshake immediately.

4.  The requester will follow the same process to construct the transcript of KEY_EX-CHANGE_RSP for MAC and verify the MAC ($mac_r$) with the ephemeral finish key ($efk$). If the MAC verification fails, then the initiator will terminate the session handshake immediately.

5.  At this point, the initiator starts preparing the FINISH message ($finish1_i$) to close the handshake.

6.  If mutual authentication is required, the initiator will create the transcript of FINISH for signing, which is the concatenation of $spdmvca$, the permanent public certificate chain or public key of the responder as identity information ($pk_r$), the KEY_EXCHANGE message from initiator ($keyex_i$) and the response generated KEY_EXCHANGER_RSP message ($keyex_r$), the permanent public certificate chain or public key of the initiator as identity information ($pk_i$), and the initiator generated FINISH message ($finish1_i$). The

initiator needs to sign the transcript with its permanent private key ($sk_i$) and generate the digital signature ($sig_i$).

7.  Then the initiator will create the transcript of `FINISH` for MAC, which is the concatenation of the transcript of `FINISH` for signing and the digital signature ($sig_i$). The initiator needs to MAC the transcript with the ephemeral finish key ($efk$) and generate the MAC ($mac_i$).

8.  The full `FINISH` message ($finishp_i$) is the concatenation of the requester generated `FINISH` message ($finish1_i$), the digital signature ($sig_i$), and the MAC ($mac_i$). The digital signature is absent if mutual authentication is not required.

9.  The final `FINISH` message ($finish_i$) is the AEAD of the full `FINISH` message ($finishp_i$) with the initiator direction handshake key ($ehk_i$).

### 6.1.4. Fourth Message

`FINISH_RSP` (responder to initiator)

Once the responder receives the `FINISH` message, it performs AEAD decryption and verifies the AEAD MAC.

1.  If mutual authentication is required, the responder needs to verify the digital signature with the initiator's permanent public key ($pk_i$). If the digital signature verification fails, then the responder will terminate the session handshake immediately.

2.  The responder needs to verify the MAC with the ephemeral finish key ($efk$). If the MAC verification fails, then the responder will terminate the session handshake immediately.

3.  As the final step, the responder creates the `FINISH_RSP` message ($finishp_r$).

4.  The final `FINISH_RSP` message ($finish_r$) is the AEAD of the responder-generated `FINISH_RSP` message ($finishp_r$) with responder direction handshake key ($ehk_r$).

After the requester receives the `FINISH_RSP`, it performs AEAD decryption and verifies the AEAD MAC. If the AEAD MAC verification passes, then the secure session is set up between the initiator and the responder. The session application message after the handshake phase is unchanged, which uses the AEAD encryption.

In hybrid mode, there will be two shared secrets, including the traditional Diffie–Hellman ephemeral (DHE) secret and the PQC shared secret. We concatenate them together as the final SPDM key exchange shared secret and input it to the SPDM key schedule algorithm.

Note that in the SPDM specification, the SPDM key schedule algorithm names the SPDM key exchange shared secret to be "DHE Secret" because the traditional algorithms only support DHE or ECDHE. We use a new term "key exchange shared secret" to avoid any misunderstanding. In traditional mode, "key exchange shared secret" is "DHE Secret". In PQC mode, "key exchange shared secret" is "PQC shared secret". In hybrid mode, "key exchange shared secret" is "the concatenation of traditional DHE secret and PQC shared secret".

### 6.2. Security Analysis

6.2.1. KEM-Based Key Agreement

In this proposal, we replace the DHE-based key agreement with a KEM-based key agreement. This mechanism is similar to the key exchange model in the TLS 1.3 hybrid design [26]. The KEM includes three parts.

*   KEM.Gen()->(esk,epk): A probabilistic key generation algorithm. It generates an ephemeral public key (epk) and an ephemeral secret key (esk).
*   KEM.Encap(epk)->(sharedkey, wrappedkey): A probabilistic encapsulation algorithm. It takes epk as input and outputs a shared secret (sharedkey) and a ciphertext pk wrapper (wrappedkey).
*   KEM.Decap(esk, wrappedkey)->sharedkey: A decapsulation algorithm. It takes esk and wrappedkey as input and outputs a shared key.

Once the shared key is calculated, the remaining steps are the same as in the existing SPDM standard, such as key derivation with the HMAC-based key derivation function (HKDF), identity authentication using a digital signature, and the proof for the owner of the session key via HMAC.

The required security property of a KEM scheme is indistinguishability under adaptive chosen ciphertext attack (IND-CCA2), which ensures security against an active attacker. All NIST PQC KEM finalists support IND-CCA2. However, two NIST PQC KEM alternate candidates, BIKE and SIDH, only support indistinguishability under a chosen plaintext attack (IND-CPA), which means they can only guarantee security against a passive attacker. The implementer should choose a proper KEM algorithm based on the security needs. In this case, we recommend ID-CCA2 schemes.

The existing SPDM only supports DHE. The DHE flow can be modeled as KEM, where

- KEM.Gen()->(esk,epk): To select an exponent $x$ and calculate $g^x$, then esk = $x$, epk = $g^x$.
- KEM.Encap(epk)->(sharedkey, wrappedkey): To select an exponent $y$ and calculate $g^y$ and $g^{x*y}$, then sharedkey = $g^{x*y}$, wrappedkey = $g^y$.
- KEM.Decap(esk, wrappedkey)->sharedkey: To compute sharedkey = $g^{x*y}$.

The details of DH-based KEM are described in RFC9180 [62].

The SPDM standard recommends that the requester and responder should generate a fresh DHE key pair for each key exchange request or response. That means some key reuse might be possible. The implementer should understand the limitation of key use, such as forward secrecy, and understand the KEM algorithm specific requirement, such as number of reuses. For example, PQC SIDH is not secure when keys are reused. We recommend always using fresh, one-time key exchange values.

### 6.2.2. Shared Secret Combiner

In hybrid mode, we calculate the *final_shared_secret* to be the concatenation of *DHE_shared_secret* and *PQC_KEM_shared_secret*, followed by the HKDF-Extract and HKDF-Expand to derive the handshake secret and data secret. See Figure 4.

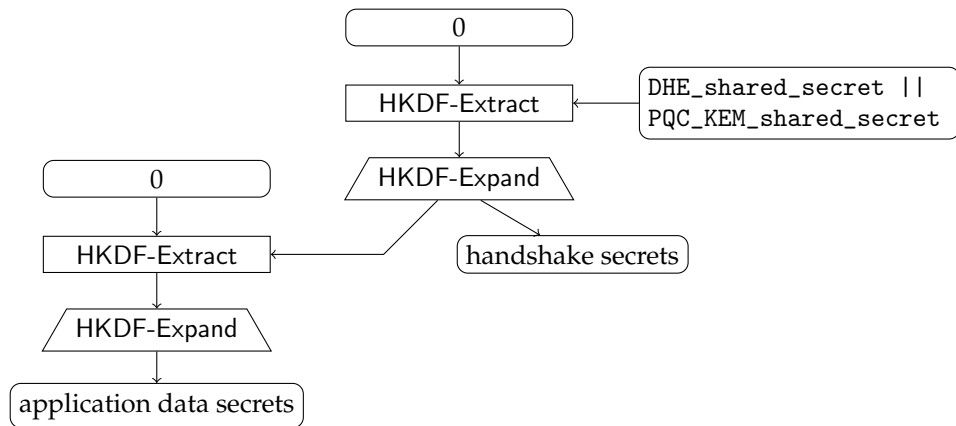

**Figure 4.** SPDM key derivation flow.

In the current SPDM and NIST PQC KEM algorithms, the length of *shared_secret* is fixed. As such, the length of *final_shared_secret* is also fixed.

The *final_shared_secret* will have the hybrid property; the secret is secure if at least one of the key exchange algorithms is secure. The analysis of KEM combiners were provided in [44,63]. The construction follows the dual-PRF combiner which was approved to be IND-CCA secure in [44].

### *6.3. Design Considerations*
6.3.1. Hybrid Key Exchange

We require that a traditional algorithm and a PQC algorithm each generate key `ExchangeData` separately and then concatenate them together. We do not pursue the option of defining a new key exchange algorithm to combine both traditional and PQC algorithms. This aligns with the current practice in TLS hybrid mode.

6.3.2. No Duplicate Message in Transport

In hybrid mode, the traditional and PQC key `ExchangeData` are combined together. There is no need to send the `KEY_EXCHANGE` message twice, namely one for classic mode and the other for PQC mode. The transcript calculation just needs to happen one time.

6.3.3. Hybrid Key Schedule

We concatenate the raw traditional DHE secret and raw PQC KEM shared secret as the final key exchange shared secret.

We do not perform any other mathematical operations, such as XOR or Key Derivation Function (KDF), to process the shared secrets. This aligns with the current practice in TLS hybrid mode.

6.3.4. Transport Message Size

As we discussed in the previous section, there is a size limitation in the transport message. The public key and the cipher text of the PQC key establishment algorithm are involved in the transmission. The public key size and the cipher text size of BIKE, HQC, Kyber, NTRU-HPS, ntrulpr, Saber, FrodoKEM, and SIDE/SIKE are larger than 256 bytes. The public key size of `classic-McEliece-{6688128,6960119,8192128}` exceeds $2^{16}$ bytes.

The current `KEY_EXCHANGE` request and response messages need to pass the key `ExchangeData`. In order to support large message transportation, we can use the generic SPDM 1.2 chunking message for such transactions.

6.3.5. Timing

As we discussed in a previous section, a PQC algorithm may have different KEM encapsulation timing requirements. Some isogenies of elliptic curves-based algorithms such as SIKE are much slower than the lattices-based or code-based algorithms, such as NTRU or Classic-McEliece. The `PQCTExponent` should be larger than the maximum encapsulation time for the supported KEM algorithms.

### 7. Results

We have developed a prototype that implements the above PQ-enabled SPDM variant based on the existing SPDM implementation [64]. We use liboqs [65] as a library that provides post-quantum algorithm implementations. The implementation is open sourced [66] which should make it relatively easy to reproduce the below tests. It includes a requester emulator and a responder emulator. The SPDM emulators can run under both the Microsoft Windows and Linux systems.

### *7.1. Test Environment*

The PQ-SPDM implementation executes in a Field Programmable Gate Array (FPGA) Smart Network Interface Card (NIC) and communicates with a host system. Both the Smart NIC and the host environment include an Intel Core CPU. We choose this high-end configuration as our first target IoT devices because we believe the high-end devices most likely would be the first ones to adopt SPDM to achieve advanced security, such as PCI express or CXL IDE for link encryption [12,13] or PCI express TDISP for confidential computing [14]. The data have been collected on an Intel Core(TM) i7-8665U CPU @ 1.90 GHz.

### 7.2. Test Method

Usually, a device is an SPDM responder, and a host operating system is an SPDM requester. We collected data from a SPDM requester and responder separately. We consider two typical use cases: (1) device authentication which includes digital signature signing and verification, and (2) one-way authenticated secure session establishment, which includes key establishment and digital signature support. In the figure, we highlight the time-consuming portions. These long-lived activities include certificate verification (CERT_-VERIFY) in `GET_CERTIFICATE`, digital signature signing (CHAL_SIGN) and verification (CHAL_VERIFY) in `CHALLENGE` for authentication, KEM generation (KEY_EX_KEM_GEN), encapsulation (KEY_EX_KEM_ENCAP), decapsulation (KEY_EX_KEM_DECAP) in `KEY_-EXCHANGE`, and digital signature signing (KEY_EX_SIGN) and verification (KEY_EX_VERIFY) in `KEY_EXCHANGE` for authentication secure session setup.

### 7.3. Test Algorithm

We collected data for the winning PQC algorithms (i.e., Kyber as a KEM and DILITHIUM, FALCON, SPHINCS+ as digital signature primitives) described in the status report of NIST PQC third round finalists [46] and compared them with the RSA and ECC results in both traditional mode and hybrid mode. For system parameters, we chose NIST security level 1, 3 and 5 separately.

Figures 5 and 6 show the data for the SPDM requester. Figures 7 and 8 show the data for the SPDM responder. L1, L3, and L5 designate the minimal NIST security level for the combination. The algorithm list used is below. Note that for hybrid modes we have both classical and PQC algorithm suites:

- L1-RSA: `ECDHE_secp256r1` + `RSASSA_3072`
- L1-EC: `ECDHE_secp256r1` + `ECDSA_NIST_P256`
- L1-RSA+KB+DL: `ECDHE_secp256r1` & `Kyber512` + `RSASSA_3072` & `Dilithium2`
- L1-EC+KB+DL: `ECDHE_secp256r1` & `Kyber512` + `ECDSA_NIST_P256` & `Dilithium2`
- L1-EC+KB90+DLA: `ECDHE_secp256r1` & `Kyber512-90s` + `ECDSA_NIST_P256` & `Dilithium2-AES`
- L1-RSA+KB+FC: `ECDHE_secp256r1` & `Kyber512` + `RSASSA_3072` & `Falcon-512`
- L1-EC+KB+FC: `ECDHE_secp256r1` & `Kyber512` + `ECDSA_NIST_P256` & `Falcon-512`
- L1-RSA+KB+SPHRK: `ECDHE_secp256r1` & `Kyber512` + `RSASSA_3072` & `SPHINCS+-Haraka-128f-robust`
- L1-RSA+KB+SPSHA: `ECDHE_secp256r1` & `Kyber512` + `RSASSA_3072` & `SPHINCS+-SHA256-128f-robust`
- L1-RSA+KB+SPSHK: `ECDHE_secp256r1` & `Kyber512` + `RSASSA_3072` & `SPHINCS+-SHAKE256-128f-robust`
- L1-EC+KB+SPHRK: `ECDHE_secp256r1` & `Kyber512` + `ECDSA_NIST_P256` & `SPHINCS+-Haraka-128f-robust`
- L1-EC+KB+SPSHA: `ECDHE_secp256r1` & `Kyber512` + `ECDSA_NIST_P256` & `SPHINCS+-SHA256-128f-robust`
- L1-EC+KB+SPSHK: `ECDHE_secp256r1` & `Kyber512` + `ECDSA_NIST_P256` & `SPHINCS+-SHAKE256-128f-robust`
- L3-EC: `ECDHE_secp384r1` + `ECDSA_NIST_P384`
- L3-EC+KB+DL: `ECDHE_secp384r1` & `Kyber768` + `ECDSA_NIST_P384` & `Dilithium3`
- L3-EC+KB90+DLA: `ECDHE_secp384r1` & `Kyber768-90s` + `ECDSA_NIST_P384` & `Dilithium3-AES`
- L3-EC+KB+FC: `ECDHE_secp384r1` & `Kyber768` + `ECDSA_NIST_P384` & `Falcon-1024`
- L5-EC: `ECDHE_secp521r1` + `ECDSA_NIST_P521`
- L5-EC+KB+DL: `ECDHE_secp521r1` & `Kyber1024` + `ECDSA_NIST_P521` & `Dilithium5`
- L5-EC+KB90+DLA: `ECDHE_secp521r1` & `Kyber1024-90s` + `ECDSA_NIST_P521` & `Dilithium5-AES`
- L5-EC+KB+FC: `ECDHE_secp521r1` & `Kyber1024` + `ECDSA_NIST_P521` & `Falcon-1024`

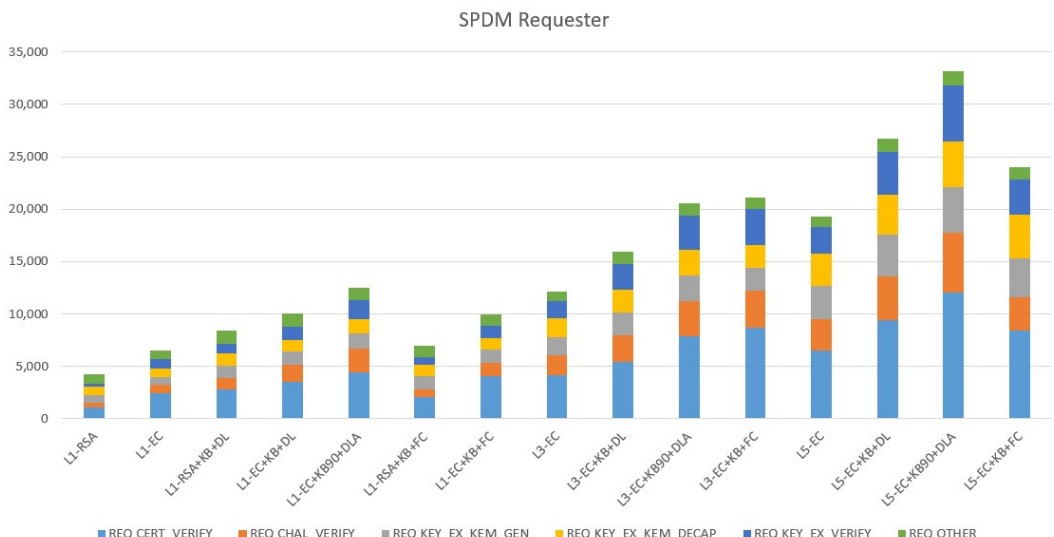

**Figure 5.** SPDM requester performance—Kyber, DILITHIUM, FALCON (microsecond).

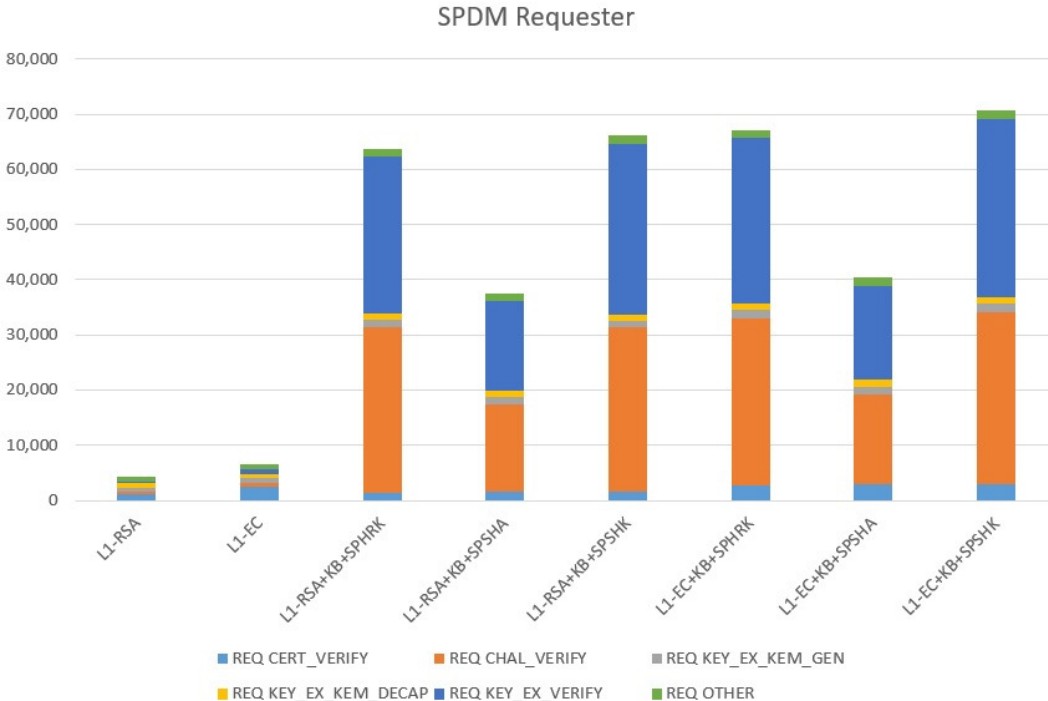

**Figure 6.** SPDM requester performance—SPHINCS+ (microsecond).

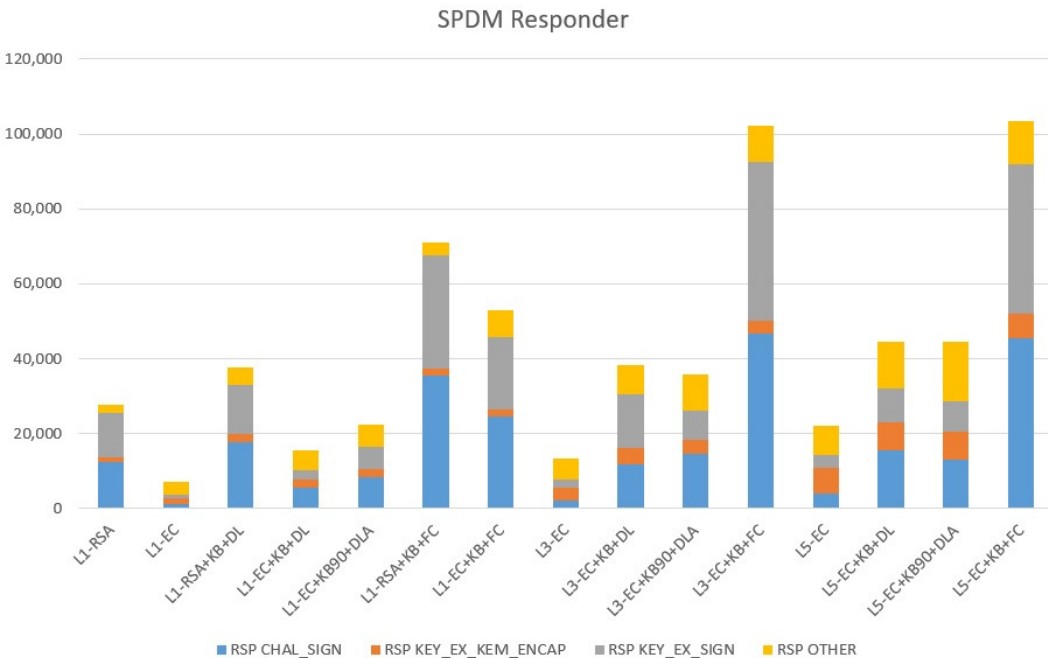

**Figure 7.** SPDM responder performance—Kyber, DILITHIUM, FALCON (microsecond).

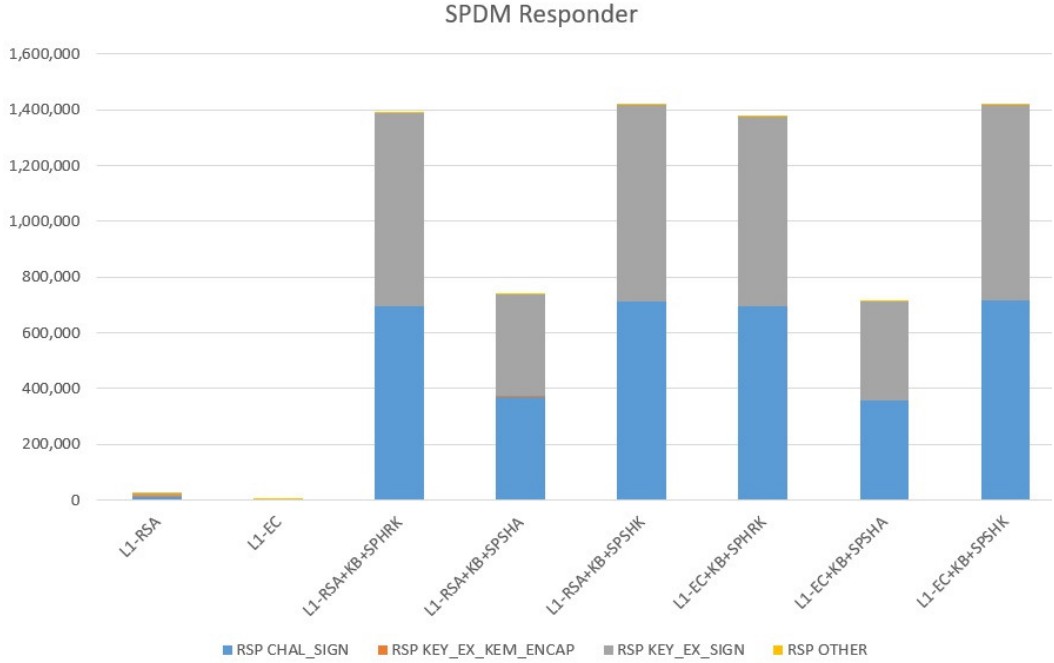

**Figure 8.** SPDM responder performance—SPHINCS+ (microsecond).

## 8. Discussion

The data on the requester side shows that the cryptography timing caused by the hybrid mode is less than double of the traditional time. Kyber, DILITHIUM, and FALCON all demonstrate good performance. The data on the responder side shows that the digital signature process contributed the majority of time. DILITHIUM is much better than FALCON. However, SPHINCS+ takes significantly longer time in both signature generation and verification.

### 8.1. Challenges

The SPDM specification defines the digital signature data or the public key `Exchange-Data` as fixed size fields. However, some PQC algorithms will generate a signature with variable size, such as FALCON. The actual size is smaller than the maximum size. The liboqs implementation uses the first four bytes to store the actual size. Care should be taken when moving the variable signature data to or from a fixed size buffer in order to avoid buffer overflows.

If there is a transport message size limitation, we have to use a chunking mechanism to transfer the large signature or public key. Chunking will increase the overhead on message disassembling and reassembling. However, even in cases where the transport layer has the capability to transfer a large message, we may still consider chunking. The rationale is that the SPDM transport layer might not be reliable. As such, a package may be lost or broken during transmission. For this scenario, a message retry may be needed. The overhead to retry a large message is bigger than the cost to retry a small chunked messaged. The implementer needs to balance the overhead of chunking against the overhead of message retry.

### 8.2. Backward Compatibility

According to the previous discussion, the PQ-SPDM design can achieve partial backward compatibility with the SPDM 1.2 protocol. See Table 3.

**Table 3.** PQ-SPDM message compatibility.

| SPDM Message | Changes for PQ-SPDM | Compatible? |
|---|---|---|
| `GET_CAPABILITIES, CAPABILITIES` | 1. Add extra capabilities (PQC, Hybrid mode) <br> 2. May add extra field `PQCTExponent` as the timing required for PQC processing | YES |
| `NEGOTIATE_ALGORITHMS, ALGORITHMS` | 1. Add PQC algorithms bits (PQC signature, PQC Key establishment) | YES |
| `GET_CERTIFICATE, CERTIFICATE` | 1. May add certificate type (PQC, Hybrid). <br> 2. May enlarge `PortionLength/RemainderLength` and `Offset/Length` field from 2 bytes to 4 bytes or retire the `CERTIFICATE` specific chunking. | NO |
| `CHALLENGE, CHALLENGE_AUTH` | 1. Add PQC or Hybrid digital signature. | YES |
| `GET_MEASUREMENTS, MEASUREMENTS` | 1. Add PQC or Hybrid digital signature. | YES |
| `KEY_EXCHANGE, KEY_EXCHANGE_RSP` | 1. Add PQC or Hybrid key exchange. <br> 2. Add PQC or Hybrid digital signature. | YES |
| `FINISH, FINISH_RSP` | 1. Add PQC or Hybrid digital signature, in mutual authentication. | YES |
| `GET_CSR, CSR` | 1. Add PQC or Hybrid digital signature. | YES |
| `SET_CERTIFICATE, CERTIFICATE` | 1. May add certificate type (PQC, Hybrid). | YES |
| Secured Message Using SPDM | 1. May enlarge `ApplicationDataLength` and `Length` field from 2 bytes to 4 bytes | NO |

### 9. Conclusions and Future Work

In this paper, we propose a way to add PQC capability to the SPDM protocol. This addition entailed some minimal modifications of the existing SPDM protocol. We successfully created a prototype based upon the proposal. This prototype shows that PQC-SPDM is feasible in practice. We hope our analysis will help prepare the industry to adopt SPDM for device firmware with long-lived service requirements. This class of solution is imperative given the looming need to have more widespread support for post-quantum security capabilities. We notice that the latency caused by the digital signature signing on the responder side might become a performance concern in the future. The KEM-based authentication [61] approach could be a way to resolve this timing problem. We will perform research on that area in the future.

**Author Contributions:** Conceptualization, J.Y. and V.Z.; methodology, J.Y. and K.M.; software, J.Y; formal analysis, J.Y. and K.M.; data curation, J.Y.; writing—original draft preparation, J.Y. and K.M.; writing—review and editing, V.Z. All authors have read and agreed to the published version of the manuscript.

**Funding:** This research received no external funding.

**Data Availability Statement:** The data collected in the paper can be reproduced by using the openspdm-pqc project [66].

**Conflicts of Interest:** The authors declare no conflict of interest.

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
