# Peer review of "Post Quantum Design in SPDM for Device Authentication and Key Establishment"

_cryptography, doi:10.3390/cryptography6040048_

Round 1
Reviewer 1 Report
This paper presents the adoption of PQC algorithms into the SPDM protocol.
All in all the paper is well written and the presentation is consistent, while the authors are active in a very relevant area of research.
I have the following concerns for the paper:
1) The paper does not have a related area section. While it may be the first one to investigate PQC algorithms for the SPDM protocol, there are many papers that deal with IoT and PQC adoption, such as:
Chung, Chia-Chin, et al. "When post-quantum cryptography meets the internet of things: an empirical study." Proceedings of the 20th Annual International Conference on Mobile Systems, Applications and Services. 2022.
The authors must discuss the most relevant papers to make clear the added value of their work.
2) While the authors mention that IoT devices are the main target of the SPDM protocol, the carried out experiments are based on Windows and Linux platforms in a virtual environment with Dual Core CPUs. How does this choice affect the numerical results and how relevant are to the IoT environment? Why not perform experiments directly in an IoT device?
3) What are the alternatives of SPDM and what is the adoption of SPDM currently?
Reviewer 2 Report
This is a very good research which has high impact to the modern world. I am satisfied with all reasonings except that more figures or diagrams can be used to illustrate the mechanism. I also agree with the authors that 'chunking' is not ideal and the derived problem is not easy to be solved flawlessly. It would be great if the authors can spend more effort in this problem in future.
Reviewer 3 Report
This paper introduces a post-quantum cryptography algorithm (PQC) based Security Protocol and Data Model (SPDM) protocol. First of all, I believe I am not in the direct research field related to this paper. After reading the paper, it seems that the paper has its contribution, but it is very hard for me to evaluate the contribution. Overall, this paper seems to be a handbook instead of a paper. There is no detailed information related to the techniques (e.g. what is the algorithm of PQC) mentioned in this paper and some title is very hard to understand. For example, the subtitle 'No Duplication' appears multiple times but the meaning of 'No Duplication' is very hard for me to understand.
In summary, I do not see this paper as a bad paper but it is very hard for me to understand and evaluate. A reader who outside this field will be very hard to understand this paper.
Round 2
Reviewer 3 Report
Thank you for the revision. It looks much more understandable now.